# Utilization of Industry 4.0 Related Equipment in Assembly Line Balancing Procedure

**Nikola Gjeldum [1,*], Bashir Salah [2] , Amanda Aljinovic [1] and Sajjad Khan [3]**

[1]   Faculty of Electrical Engineering, Mechanical Engineering and Naval Architecture (FESB), University of Split, 21000 Split, Croatia; amaljino@fesb.hr
[2]   Industrial Engineering Department, College of Engineering, King Saud University, P.O. Box 800, Riyadh 11421, Saudi Arabia; bsalah@ksu.edu.sa
[3]   Supervising Electronics Engineer, Saudi BinLadin Group, P.O. Box 6807, Ibrahim Amin Fudah St., AR Rawdah District, Jeddah 23434, Saudi Arabia; sajjadkhanms@gmail.com
*   Correspondence: ngjeldum@fesb.hr

**Abstract:** In recent decades, production in high-volume/low-variety batches is replaced with low-volume/high-variety production type. This type of production demands excessive flows of both material and information. Recent advances in information and communication technologies (ICT), together with the concept of cyber-psychical system (CPS) enable the concept of Industry 4.0 (I4.0). In this paper, the performance of I4.0 related equipment implementation is presented in iterative assembly line balancing (ALB) process of a gearbox assembly line. Largest candidate rule method through spreadsheet simulation was used for tasks reallocations, with the objective to minimize the cycle time when the number of stations is fixed. Utilization of human analysts using snap back method for manual data gathering process still shown advantage over I4.0 equipment utilization in manual ALB. The assembly process is performed in the learning factory environment, and it is considered as very close to real industry process. The major conclusion is that I4.0 is excellent in process data monitoring and product tracking, but activities to be performed to effectively exploit I4.0 is demanding for task reallocations during the balancing procedure. Nevertheless, future enhancements of I4.0 system are listed to bridge this gap and to increase I4.0 system usefulness in the manual assembly line balancing process.

**Keywords:** assembly line; assembly line balancing; Industry 4.0 related equipment; production variety; information and communication technology

## 1. Introduction

Assembly lines (ALs) consists of successive workstations connected by a material handling system. On workstations (WS), employees carry out a defined proportion of an assembly work needed to complete a subassembly or final product of a predefined structure within a set time span. The ALs were first introduced in the 1913s by Henry Ford to meet the customer demands and expectations [1]. They contribute to increased productivity and lower cost by limiting both, single worker per products and nonvalue-added activities related to movements and manipulations. The production activities are allocated to WSs along with a set of preceding activities to perform a task. However, in today's turbulent and globalized market, it is sometimes insufficient to just meet the customer demands and expectations [2]. ALs contribute to facing continuous changes in a company's internal and external environments, due to several factors, including the challenges of meeting the fluctuating production volumes/varieties, balancing the assembly line, improving production flow layout, etc.

The manual assembly process is a sequence of successive and parallel activities performed by humans, to achieve a subassembly or final product of a predefined structure. The sequence is predominantly determined by the product design, in general, and the product structure, in particular [3]. Manual assembly is typically used for prototype and low-volume work, whereas the postautomated assembly is used for odd-form components [4]. These processes are complicated in terms of their design and development and are difficult to realize theoretically [5]. Manual assembly involves the composition of previously manufactured components and/or subassemblies into a complete product or unit of a product, primarily performed by human operators using their inherent dexterity, skill, and judgment [6]. Consequently, with the increase of customized built-in elements of the assembled product, the complexity of the assembly processes will be increased. In the last two decades, a considerable amount of researches in operation management have focused on several models and techniques for assigning tasks to workstations to optimize a specific objective function in order to satisfy several constraints [7].

Many previous publications discussed the issues related to the design and operation of ALs in order to optimize their performance [8,9]. These issues include concerns about high-volume/low-variety and low-volume/high-variety production types of ALs, i.e., try to reveal how to determine settings and trade-offs related to the number, type, and order of product variant to be produced. The shapes of the ALs can be divided into the following classes [2,10,11]:

- Single-product ALs
- Multiproduct ALs

  o　　Multi-model production, i.e., batch production and/or mass production
  o　　Mixed-model production, i.e., product variety.

Assembly line balancing (ALB) aims to dedicate the activities to the successive workstations so that workloads are equally distributed among the workstation [12]. The most commonly investigated assembly line balancing (ALB) is the simple assembly line balancing (SALB) [13]. Besides other assumptions listed in [14], the SALB problem is defined by production of one standardized product, with common cycle time, tasks sequence subjected to precedence constraints, deterministic task times, each task assigned to only one workstation, and all stations equally equipped. As addition to SALB's simplifying hypotheses, general assembly line balancing problem (GALB) studied different aspects of an assembly line [15]. Multiple researchers have studied different GALB problems [16], in terms of product diversity introduction [17,18], allocation of workers according to their skills [19], and consideration of ergonomic restrictions and workers' travel [20,21]. Lines with nondeterministic task times are studied in [22].

Two basic metrics that need to be tracked during the designing of the low-volume/high-variety assembly line are how to maximize the workers' utilization and how to minimize lead time. For low-volume/high-variety production line, if organized as one-piece flow, it is possible to achieve the increase of workers' utilization up to some limit by the balancing process. If work-in-process (WIP) is allowed, further increase of workers' utilization could be achieved by the balancing process, but with the increase of lead time. In asynchronous assembly lines, downstream starvations and upstream blockages can occur. Buffers or WIP are often employed to minimize these problems, which are particularly relevant when the line is shared between a set of different products models [23]. Increased inventories at buffers along assembly line could increase workers' utilization, but excessive balancing and scheduling should be performed. In the case of sequence-dependent setup times occurrence [2], binary linear mathematical programming (BLP) model and a simulated annealing (SA) algorithm are proposed to model and solve balancing and scheduling tasks. In reference [24], a mixed-integer programming (MIP) mathematical model is used to formulate mixed-model two-sided assembly lines problem. An effective variable neighborhood search (VNS) algorithm is proposed to solve the balancing problem, especially for the large-sized problems. Metaheuristic solution approaches

(nonexact solution approaches) has to be used when the problem size is increased [25]. Simulated annealing algorithm [26,27], ant colony optimization [28], genetic algorithm [29,30], and many other approaches have been developed and found in literature.

Traditional ALs are no longer suitable for today's industrial environments. A new generation of assembly systems is needed to take advantage of numerous opportunities that the fourth industrial revolution (I4.0) offers, such as higher productivity, enhanced flexibility, agility, tracking, and traceability [7]. In the case of ALs, Y. Cohen et al. [9] recently investigated breakthrough solutions that take advantage of the I4.0 where a general architecture was provided to implement I4.0 enablers into existing systems. All different aspects in assembly line, where GALB should be used, increase the need for comprehensive information flow. Models presented by researchers present dynamic sequencing to confront low-volume/high-variety production requirements. However, those dynamic approaches impact on workers' exertion due to constant monitoring of excessive information arrivals, while performing assembly tasks. Addition of time to act according to available information, the constantly changing production plan in order to increase workers' utilization, additionally burdens workers. On the other side, if I4.0 related equipment is set up intuitively, it could reduce the amount of tasks related to information flow performed by the worker compared to conventional information flow on paper sheets. Sensors for tracking, e.g., radio-frequency identification (RFID) technology, are utilized for real-time data gathering, while displays on workstations offer instant insight in the information needed in a particular moment. Pushbuttons, keyboards, or touch screens improve feedback information flow.

One of the most significant technologies for automatic identification and tracking entities in the production and service systems that provides precise real-time information about the observed products is RFID technology [31]. RFID systems are one of the key technologies in the Internet of Things (IoT), which empowered manufacturing sites [32]. RFID is recently used in high complexity ALs for data processing, especially under uncertainty [33]. This technology is based on radio-frequency waves used to transfer data between RFID readers and RFID tags. An RFID reader is connected to a computer, microcomputer, or with the microcontroller as a processing device. RFID tags are available in different shapes and sizes, and they consist of an antenna and a small chip that carry data. In read mode, the RFID reader detects an RFID tag and records the data stored to transfer those to the computer for processing [34]. In write mode, RFID reader/writer is used to record strings of data on RFID tag. For both modes, the tag has to pass contactless within the reader's operating range with speed slow enough to enable the unambiguous success of reading or writing actions [35].

Adaptation to I4.0 system is commonly perceived as stressful to workers. Some level of evasion for education about system usage can be expected. The I4.0 introduction should be done in a way that employees gain clear insight in usefulness and advantages of those systems, with the premise that those systems will hardly be misused in terms of work ethics.

This paper reveals the usefulness of I4.0 related equipment utilization compared to traditional information flow on paper sheets, from two aspects: usefulness of I4.0 related equipment utilization in assembly balancing procedure and usefulness of I4.0 related equipment utilization in low-volume/high-variety assembly line.

In this paper, two information flow systems supporting the ALB process are compared. The first system incorporates manual data gathering process and is referred to as "Manual approach." Paper instructions to workers and a paper spreadsheet with production sequence available at every workstation presents well-known conventional information flow. The second one incorporates I4.0 related equipment for automatic data gathering process and is referred to as "I4.0 approach." Instructions are listed on touch Liquid Crystal Display (LCD) screens according to production sequence and automatically presented information about which gearbox type is currently processed on respective WS.

The remainder of this paper has the following structure: gearbox assembly process setup is presented in Section 2. Problem definition and procedure diagram for ALB are also presented together

with results of three ALB iterations. In Section 3, the discussion is done. The focus is set to I4.0 related equipment utilization in ALB problem solving. Finally, conclusions are presented in Section 4.

## 2. Gearbox Assembly Line Balancing Procedure

### 2.1. Gearbox Assembly Process Setup

A car transmission, i.e., manual gearbox, was chosen for the case study due to its complexity and the relatively large number of parts. Due to complex mechanical parts, like gearboxes parts, manual assembly is still very ubiquitous in the automotive industry [36]. Assembly of such parts can be automated with excessive investment, due to a complex assembly process, and some attempts with the automation approach failed with the conclusion that humans are underrated [37].

The gearbox is a mechanical device that transfers torque from the car engine to vehicle wheels. It is used both to transform torque and horsepower to various speed limits and to enable the reverse drive. It is made up of a large number of different size gears and other parts such as gear levers, counters, shafts, bearings, screws, etc. In manual gearbox, different gears are engaged by the driver's shifting stick shift, which results in transmission ratios changing.

The case study presented in this paper is a gearbox assembly line, which is installed in the learning factory (LF). Learning factories are developed within university or research institution laboratories to simulate environment as close as possible to the industrial environment [38]. The gearbox assembly line was developed as a part of the lean learning factory (LLF) at Faculty of Electrical Engineering, Mechanical Engineering and Naval Architecture at the University of Split, Croatia [39]. The assembly line, containing real hand tools, supermarkets, and conveyer established to assemble a real product, the gearbox, could be considered as an industry-relevant case study of sufficient extent, thus avoiding the necessity for abstraction, assumption, and simulation [40].

The starting point for this paper investigation is presented in the authors' previous work [41]. For set-up presented in [41], the assembly line was balanced. The assembly process is performed on a total of five workstations (WS1 to WS5). Only one gearbox product type was assembled in [41] and mere information flow, on paper sheets, was adequate. Another feature was manual screwing processes on all WSs. The assembly process had high process time variations [41], especially on WSs with numerous parts to be installed, especially screws and nuts. The process times were contingent on workers' skills and adroitness while screwing a large number of screws manually.

For this paper investigation, to introduce low-volume/high-variety production, a total of six different product types of gearboxes are assembled (Figure 1), i.e., YAS, YZS, SAN, SZS, SAS, and SZS type. Gearbox casings type defines the first letter of type designation (Y or S), differences in synchronization devices define the second letter (A or Z), and mounting parts embedded define the third, last letter in type designation (S or N). Although gearboxes of different product types consist of an approximately similar number of parts and have to be assembled on the same number of WSs, differences affect information flow and increase the number of necessary parts storages on WSs, i.e., supermarkets. Additional activities are added to workers, i.e., reading the production plan on a regular basis and using instructions to select appropriate parts to be installed in every gearbox.

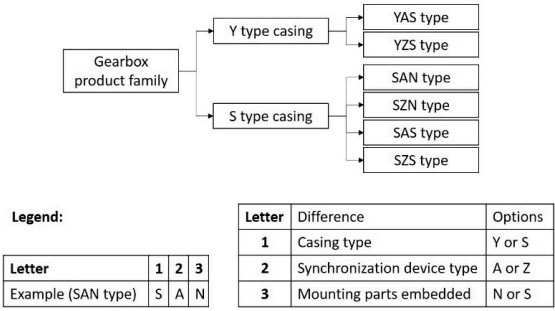

**Figure 1.** Six different gearbox product types.

For the balancing process, assembly preceding diagram for the first balancing iteration is developed. The assembly diagram is a series of assembly actions which can be shown in the form of a large variety of diagrams. The assembly diagram adapted to a large number of gearbox parts is shown for SAN type gearbox in Figure 2.

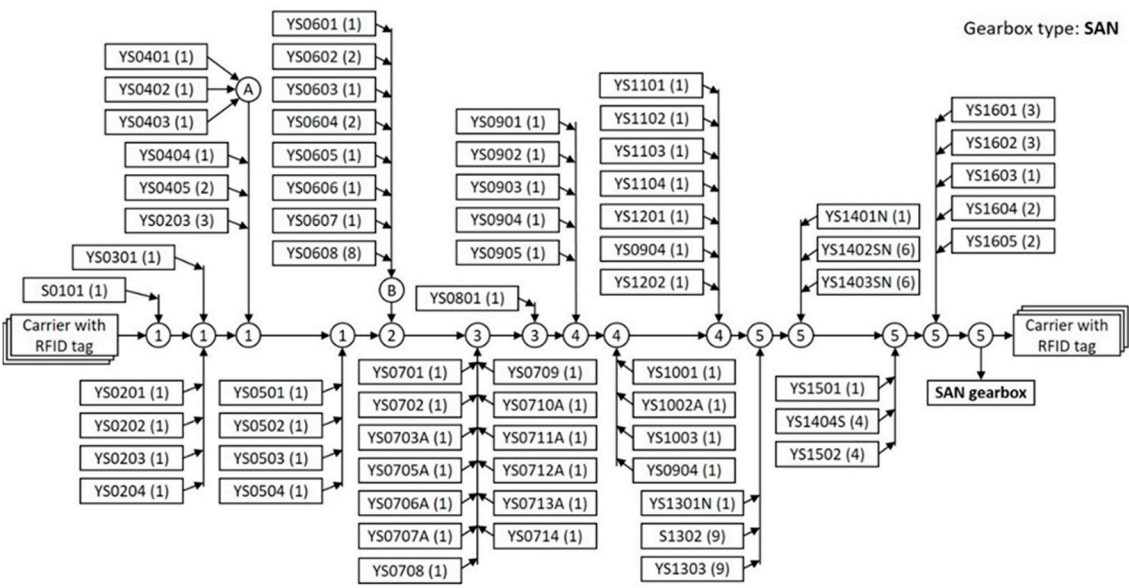

**Figure 2.** Assembly diagram for SAN gearbox type (RFID—radio-frequency identification).

As there are differences in parts, a total of six assembly diagrams are developed for six different product types. In further considerations and explanations, SAN gearbox diagram and parts (Figure 2) will be used. In SAN gearbox, letter "S" in "S0101" part (casing) defines the first letter of SAN designation. Parts with designations ending with "A" defines the second letter, while designation ending with "N" defines the third letter in SAN designation. The parts starting with "YS" and ending with four numbers are universal parts and has to be inserted in every gearbox type.

Assembly steps assume an insertion of individual part or group of parts that presents one entirety. The first step in the assembly process is positioning of the carrier that is suitable for rolling over conveyer on WS1, succeeded with base casing positioning (part S0101) on the carrier. Succeeding parts which is installed in WS1 are YS0201 followed by YS0202, YS0203, and YS0204. Assembly is performed in the second assembly step. The second assembly step can be performed in parallel with the insertion of YS0301. Therefore, the insertion of YS0301 could be also assumed as the second assembly step. Both steps are pointed to one circle of an assembly process diagram, marked with 1. The number of WS in which assembly of parts or assembly steps are performed is shown in circles in Figure 2. An instruction page prepared for assembly of the aforementioned steps on WS1 is shown in Figure 3. Up to three possible sequence of picking parts from supermarket storage boxes and insertion in gearbox are shown to workers under "Possible relative order of insertion".

The circles with letters A and B in Figure 2 present the subassemblies which have to be assembled prior to insertion in the gearbox. The minimum number of subassemblies is two, and the number of more subassemblies can be assembled prior to insertion in the gearbox. It is found that major differences in assembly steps sequences could also be introduced. For example, assembly step including parts YS1101, YS1102, and YS1103 can be inserted in gearbox already in WS1. The assembly line balancing procedure presented in this paper takes all possibilities in tasks reallocations processes.

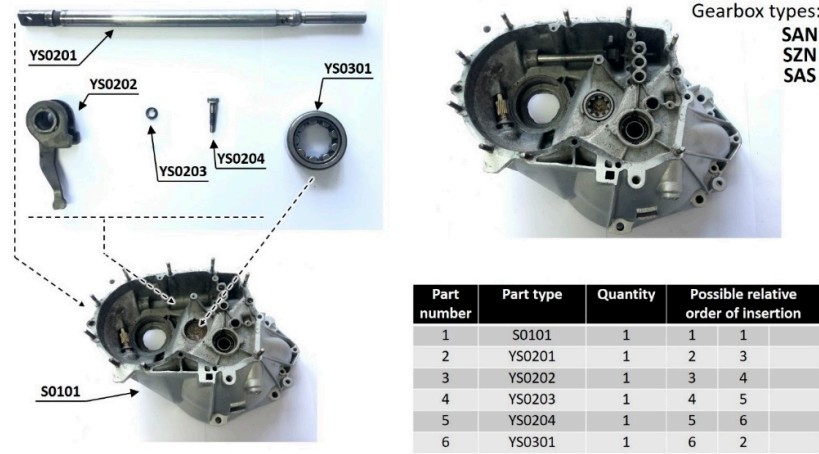

| Part number | Part type | Quantity | Possible relative order of insertion | |
|---|---|---|---|---|
| 1 | S0101 | 1 | 1 | 1 |
| 2 | YS0201 | 1 | 2 | 3 |
| 3 | YS0202 | 1 | 3 | 4 |
| 4 | YS0203 | 1 | 4 | 5 |
| 5 | YS0204 | 1 | 5 | 6 |
| 6 | YS0301 | 1 | 6 | 2 |

**Figure 3.** Instruction page for two assembly steps of SAN gearbox type.

## 2.2. Assembly Line Balancing Procedure

In Figure 4, the assembly line balancing procedure used in this paper is presented.

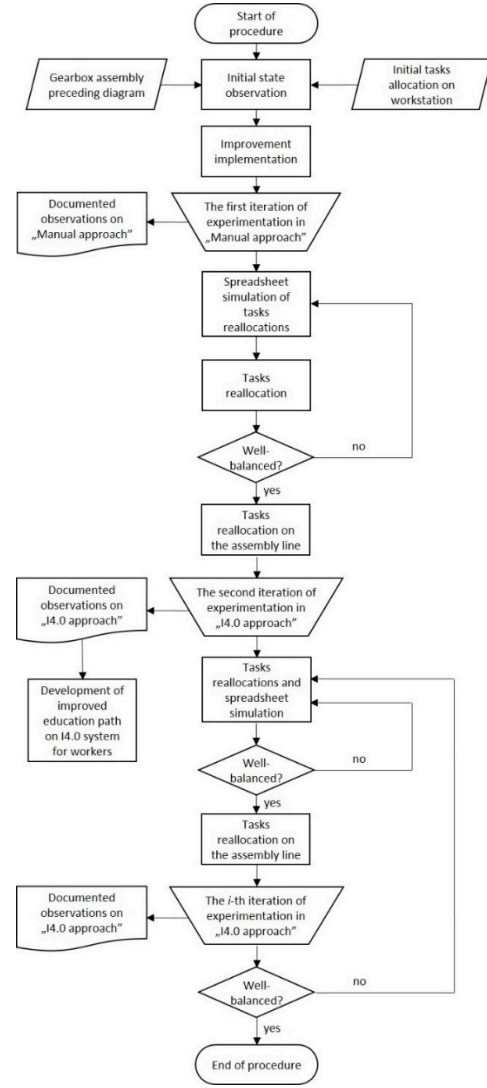

**Figure 4.** Assembly line balancing procedure.

The assembly line balancing procedure is performed in a number of iterations (*i*) sufficient enough to achieve satisfying workload balance among WSs. Two different data gathering approaches are used: "Manual approach" and "I4.0 approach" (Figure 4). The one part of the gearbox assembly line set-up, prepared for the first iteration of experimentation, is presented in Figure 5.

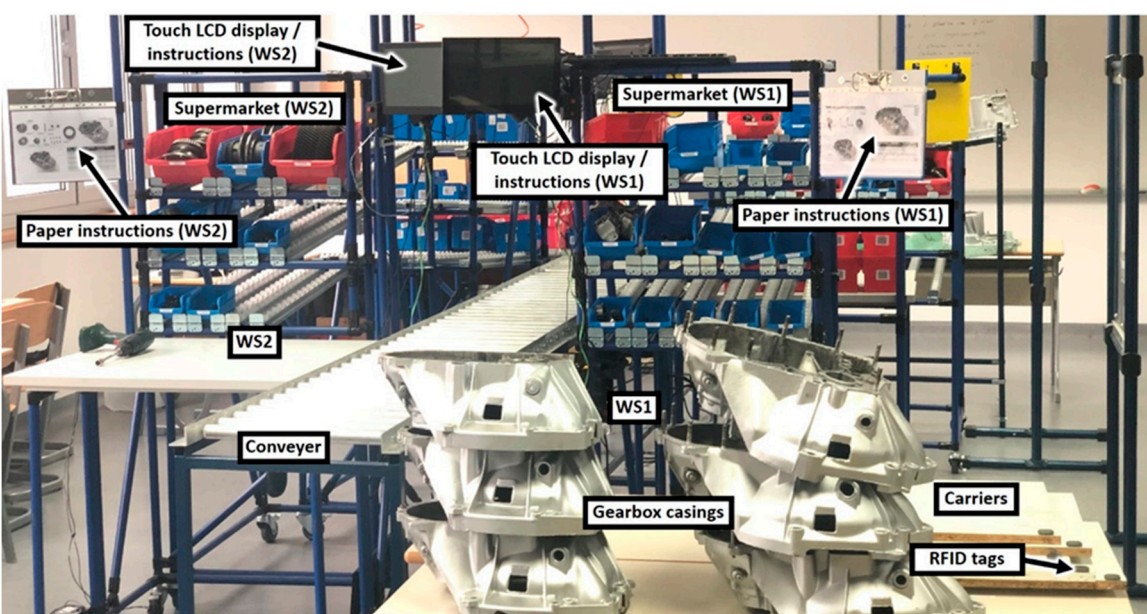

**Figure 5.** Gearbox assembly line setup for the first iteration (LCD—Liquid Crystal Display, WS1—workstation 1, WS2—workstation 2).

### 2.3. Results

#### 2.3.1. Manual Approach for the First Iteration

The starting point of the procedure is the observation of the current state presented in previous work [41]. The process times and process times standard deviations are shown in Table 1.

**Table 1.** Process times and standard deviation in initial state (WS3—workstation 3, WS4—workstation 4, WS5—workstation 5).

|  | WS1 | WS2 | WS3 | WS4 | WS5 |
|---|---|---|---|---|---|
| **Average process time $t_{e,i}$ (s)** | 297.00 | 310.20 | 140.80 | 335.70 | 214.00 |
| **Standard deviation of process time $\sigma_{te,i}$ (s)** | 34.19 | 51.70 | 25.13 | 33.84 | 43.78 |

Prior to the first iteration of assembly line balancing procedure, improvements on setup presented in [41] are introduced. Excessive workloads on WS1, WS2, WS4, and WS5, compared to WS3, are attempted to be solved by the introduction of electric screwdrivers and improved instructions to workers. WS5 is additionally equipped with gravity conveyer to relief worker from lifting heavy upper casing to compensate time needed for 11 new parts to be assembled on that WS. WS3, as the WS with the lowest process time, is significantly influenced by the introduction of low-volume/high-variety production, in terms of many different parts to be inserted according to product type. That should increase the process time on WS3. Therefore, it is expected that the assembly line will be better balanced with the implementation of improvements and changes in a number of parts compared to the case shown in [41].

Assembly process with "Manual approach" used information flow on paper sheets and manual data gathering process by analysts. Ten participants are required to perform experimentation run.

Five participants are workers, working on five WSs, and another five participants are trained time study analysts, as every WS requires one to achieve realistic process time measurements and additional observations. Analysts' first task was to recognize feasible assembly steps. The warm-up period prior to experimentation enabled analysts, to define assembly steps, and workers, to achieve the necessary skill in performing assembly. During experimentation run, analysts gathered process times of every individual assembly step. Total process time on every WS was calculated by the summation of assembly steps process times. Additional analysts' task was to monitor workers' activities, related to the usage of the production plan and to the usage of available work instructions on paper sheets.

Production plan was available on every WS in the form of paper spreadsheet revealing assembly sequence. Detailed paper instructions, including photographs of parts to be installed, were hanging in sight of workers, on paper sheets. As there was more than one assembly step defined on every WS, and there are differences between types of gearboxes, eight paper sheets in average were available for listing at every WS. The first iteration of experiments was performed with a total of 24 gearboxes assembled.

Method of timing by stop watch, i.e., snap back method [42] is used for time gathering purpose. Abnormal values during readings are recorded together with its causes of abnormality. Those values can be used in further analysis to improve processes. If those abnormalities occurrences are neglected, process times variations increase abnormally, which do not reflect the real process times [42]. The simplified method used includes only three causes of abnormality: the problem with tools, denoted as "T," the problem with parts, denoted as "Z," and bad material detected, denoted as "M." Detection of abnormalities and duration of those were important in further considerations for line balancing and for assembly improvement process on every WS.

After the first iteration experimentation run, the manually written data on snap back method template were transferred to digital spreadsheet in order to calculate mean and standard deviation of process times automatically. Table 2 shows data gathered for all five WSs. Average process times with error bars presenting standard deviations are shown in Figure 6.

**Table 2.** Experimentation results for the first iteration.

|  | WS1 | WS2 | WS3 | WS4 | WS5 |
|---|---|---|---|---|---|
| **Average process time $t_{e,i}$ (s)** | 263.72 | 220.78 | 129.40 | 344.82 | 352.35 |
| **Standard deviation of process time $\sigma_{te,i}$ (s)** | 32.43 | 39.15 | 19.76 | 52.93 | 26.95 |
| **Abnormality Z occurrences** | 2 | 4 | 2 | 1 | 0 |
| **Abnormality Z average time (s)** | 45.23 | 86.88 | 16.30 | 180.12 | – |
| **Abnormality Z occurrences** | 3 | 1 | 0 | 1 | 2 |
| **Abnormality T average time (s)** | 23.11 | 36.87 | – | 67.48 | 22.31 |

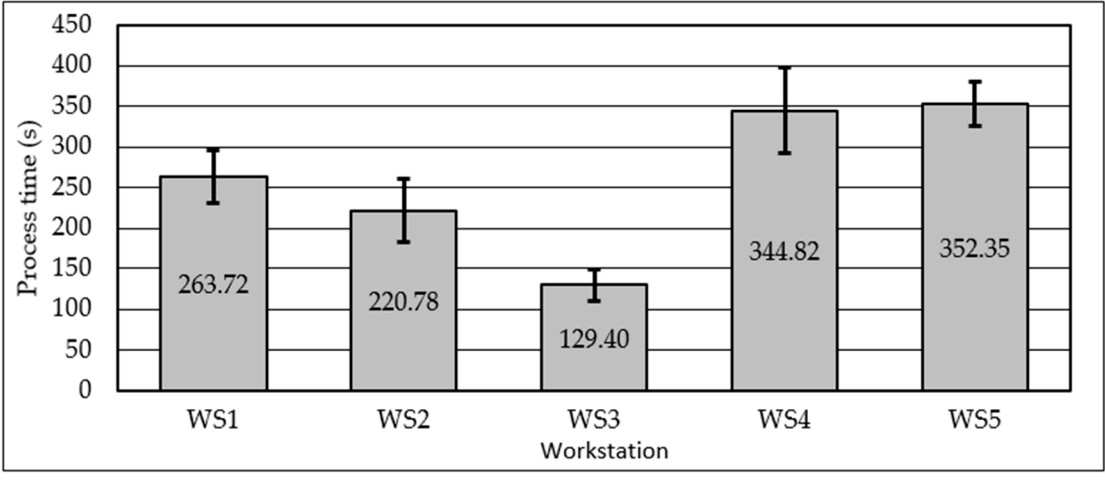

**Figure 6.** Process times and standard deviations in the first experimentation iteration.

It can be seen in Figure 6 that process times vary enormously among WSs. Achieved results showed that workload distribution, as a result of the previous investigation [41], was not appropriate, although implemented improvements and introduction of product variety were expected to improve work balance among WSs. During observations, another issue disbalancing assembly line was found. It laid in the mutual impact of one WS activity on neighbor WS activity. During assembly, bottleneck occurred on WS5. There are WSs with higher workloads than neighboring WSs (i.e., WS1)." When WS is a bottleneck or have a higher workload, there is a starvation, i.e., lower utilization on succeeding WS, and idle time occurs. During this idle time, i.e., waiting time for the next product, worker can perform some activities and some improvements. Assembly steps are generally planned to be performed when product arrives at WS. However, the activities that can be done in idle time, without the presence of gearbox are: introduction with the following product type, determination of parts location needed for the following product, selection and pull-out of parts from supermarkets and disposition on the worktable, the formation of subassembly on worktable although those are not defined in preceding diagram, etc. Therefore, WS with shorter process time will reduce its process time even more, as it can work in parallel with preceding WS. The analyst starts to gather process time as instructed, at the moment when gearbox arrives on WS. Additional activities, done prior to gearbox arrival are partly recorded, but cannot be assigned easily to some activity or to some product. It is the case especially if preparation and improvement is done to next few products. So, this data cannot be analyzed or used subsequently in the line balancing process.

This phenomenon is neglected in manual assembly lines where the arrival of the product on WS is the trigger for the start of process time gathering. Human workers are more flexible and intelligent then programmed robots, so the human worker will adapt to the situation and improve their activities. Although standard work procedure is deployed for all WSs, it is in human nature to improve processes and speed it up if found possible. This is making assembly line even worse balanced in this case (Figure 6). By assembly process observation, it was found that worker on WS3 utilized idle time in collecting parts YS0701 to YS0714 on the worktable, in disposition appropriate for fast installing when gearbox arrived on WS3. In later experimentation period, parts YS0701 to YS0714 were assembled in subassembly on the worktable and inserted in gearbox in a very short time (about 4 s needed). This reorganization and improvements initiated by the worker on WS3 enabled even higher disbalance than found in previous research [41].

A total of 22 assembly steps were recognized and individual process times were collected. Those assembly steps were used for reallocation purposes in following balancing processes. The commonly used balancing method, largest candidate rule (LCR) [43], was used. The goal was to find the minimum number of WSs for defined customer overall cycle time, i.e., assembly line overall cycle time. In the presented case study, the number of WSs was predefined, by layout constraints, as five. The objective was minimizing the cycle time when the number of stations was fixed, and it was considered as "Type 2" problem, according to [44]. Therefore, assembly steps were reallocated only among existing WSs, taking the preceding diagram into consideration. Prior to that activity, the total assembly process time was used in order to the get ideal overall cycle time, $\bar{c}_t$, after 1 iteration according to the Equation (1):

$$\bar{c}_t = \frac{\sum\limits_{i=1}^{n} t_{e,i}}{N_{WS}} = \frac{1311.07}{5} = 262.21 \text{ (s)}, \tag{1}$$

where $N_{WS}$ is the number of workstations.

Ideal overall cycle time is the process time goal for every WS. As the total number of WSs is limited to five, achievable overall cycle time found in the bottleneck process is higher than the ideal overall cycle time. Therefore, the balancing process is performed by spreadsheet simulation according to LCR in the sufficient number of iterations. As minimum assembly step duration was 8.12 s, rounded increase of 5 s in every iteration on the ideal overall cycle time was appropriate. In iteration when assembly line overall cycle time was increased to 285 s, all assembly steps were allocated in only five

WSs and that was considered as successfully performed balancing process. Only four assembly steps reallocations were found. Table 3 presents reallocation of assembly steps.

**Table 3.** Reallocation of assembly steps in balancing procedure after the first iteration.

| | | WS1 | WS2 | WS3 | WS4 | WS5 |
|---|---|---|---|---|---|---|
| **The first iteration $t_{e,i}$ (s)** | | 263.72 | 220.78 | 129.40 | 344.82 | 352.35 |
| 1. Task reallocation | Description: YS1301N, YS1303, and S1302 from WS5 to WS4 | - | - | - | +122.26 | −122.26 |
| 2. Task reallocation | Description: YS1001, YS1003, and YS0904(1) from WS4 to WS2 | - | +62.17 | - | −62.17 | - |
| 3. Task reallocation | Description: YS0901, YS0902, YS0903, and YS0904 from WS4 to WS3 | - | - | +97.83 | −97.83 | - |
| 4. Task reallocation | Description: YS1002A and YS0904(1) from WS4 to WS3 | - | - | +38.24 | −38.24 | - |
| **Predicted $t_{e,i}$ (s)** | | 263.72 | 282.95 | 265.47 | 268.84 | 230.09 |
| **Predicted deviation from $\bar{c}_t$ (s)** | | 1.51 | 20.74 | 3.26 | 6.63 | −32.12 |

It was found that the optimal distribution resulted with the highest process time (bottleneck process time) of 28,295 s on WS2. Therefore, WS2 was expected to be a bottleneck process in the second experimentation iteration.

The expected work distribution was optimal when using 22 assembly steps. However, it is still not perfectly balanced, as shown in Table 3. The assembly step represented a part or a group of parts to be inserted in the gearbox to build some feature or entirely. Further fragmentation of assembly steps was not appropriate, as fragmentation generally increased the number of different parts on supermarkets and the complexity of the assembly process on most WSs. For example, switching installation and screwing process for five sets (from total nine) of washers and nuts (YS1303 and S1302) from WS4 to WS5 will improve balancing to some extent. However, two more part boxes should be installed in the supermarket of WS5, and clear guidance to install only four sets of washers and nuts on WS4 should be emphasized. Problems that also can be expected comprise unnecessary doubling of tools, missing to install some washers and nuts, missing to tighten all nuts to final torque, more intensive information flow resulting with an increased number of instruction pages, etc. Therefore, only a total of 22 assembly steps remained for reallocation purposes during the balancing process.

Table 3 shows predicted process times on WSs according to spreadsheet simulation. When reallocating assembly steps to another WS, its process time could be changed due to more reasons. The WS and supermarket layout and the instructions were changed. Level of gearbox completion could also be changed, which, therefore, changing accessibility when installing parts, thus increasing or reducing the speed of insertion. Therefore, the second iteration of experimentation should validate spreadsheet simulation results.

After the reorganization of the assembly line, the introduction of I4.0 related equipment and update of instructions pages, the second iteration of experimentation run took place. In this experimentation run, "I4.0 approach" was used for information flow and data gathering process.

2.3.2. I4.0 Approach for the Second Iteration

For transportation of the gearboxes along the assembly line, the carriers were used. Each carrier had glued passive RFID tag that carried the information which was not related to any of gearboxes. RFID system installed on every WS was controlled by Raspberry Pi 3B+, manufactured by Sony UK Technology Centre, Pencoed, UK, (RPi) server with Raspbian GNU/Linux 9.6, manufactured by Raspberry Pi Foundation, Cambridge, UK, over a Wi-Fi connection. On every WS, there was Raspberry Pi 3B+ microcomputer with touch LCD display and μFR Nano RFID reader with an antenna. The system was programmed in "Node.js" version 11.6.0, manufactured by OpenJS Foundation,

Joyent, Inc., San Francisco, United States of America, with execution environment using "Express.js", framework, manufactured by OpenJS Foundation, Joyent, Inc., San Francisco, United States of America. JavaScript, manufactured by OpenJS Foundation, Joyent, Inc., San Francisco, United States of America was used with "HTML" supporting "CSS", manufactured by Tim Berners-Lee contractor of *Organisation européenne pour la recherche nucléaire* (CERN), Geneva, Switzerland, for the graphical user interface. Additional push buttons, for process completion signal and for the listing of instructions were installed in the separate electronic box positioned near to LCD display. The touch LCD display displayed the instructions to workers, same to those found on paper sheets. Another functionality found on display, besides time counters, was the production plan table line defined by the type of product to be assembled. There was another table line with following product to be assembled and field that revealed wait time, if the product was already completed on preceding workstation.

Additional buttons, namely, T, Z, M, and Normal gave the possibility to worker to switch counters if the process was abnormal and then switching back to normal when the issue was solved. With the STOP button, counters could be stopped and resumed at any moment, if there was a need for leaving the working place.

The server read the regularly updated spreadsheet document which was the production plan. When an RFID tag passed by the antenna on WS1, the server sent information which product was to be assembled and assigned text string from RFID tag to the product. By tracking this RFID tag, the server monitored the location and gathered the process times and abnormality times on every WS for that particular product. When the product was completed on WS5, server released RFID tag information and calculated the total lead time for the specific product. The carrier with the RFID tag could be used again for another gearbox, without any action required. All collected data about the product assembling was available on the server for postprocessing purposes. The main components of I4.0 RFID system and display appearances are shown in Figure 7.

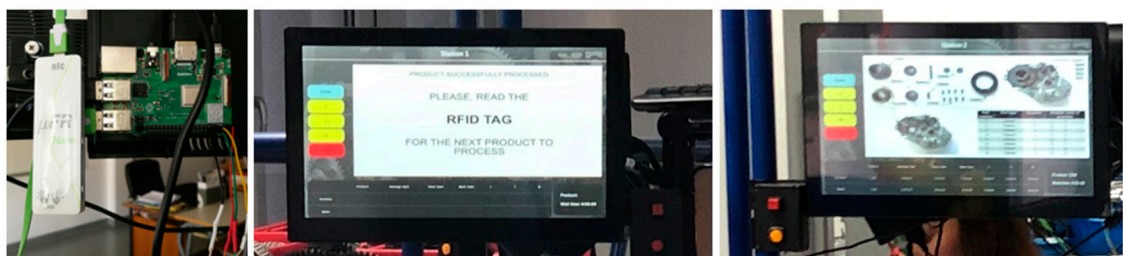

**Figure 7.** Radio-frequency identification (RFID) antenna, Raspberry Pi 3B+ (RPi), pushbuttons, and different appearances of touch Liquid Crystal Display (LCD).

Only five workers were necessary for performing experimentation in the second iteration, as I4.0 related equipment introduction aims to replace analysts. Nevertheless, analysts were present to gather data about workers' behavior on performing activities according to deployed procedures and instructions. Control stopwatches were used by analysts, to gather cycle times on WS, although "I4.0 approach" did not require analysts with stopwatches. Total 24 gearboxes were assembled. The data gathered by I4.0 related equipment are listed in Table 4. Average process times with error bars presenting standard deviations are shown in Figure 8.

**Table 4.** Experimentation results from the second iteration.

|  | WS1 | WS2 | WS3 | WS4 | WS5 |
|---|---|---|---|---|---|
| Average process time $t_{e,i}$ (s) | 211.64 | 177.60 | 309.48 | 251.99 | 229.23 |
| Standard deviation of process time $\sigma_{te,i}$ (s) | 22.31 | 27.12 | 46.13 | 22.87 | 19.96 |
| Abnormality Z occurrences | 1 | 2 | 4 | 0 | 0 |
| Abnormality Z average time (s) | 28.37 | 104.12 | 26.78 | - | - |
| Abnormality T occurrences | 0 | 0 | 3 | 0 | 4 |
| Abnormality T average time (s) | - | - | 21.14 | - | 29.44 |

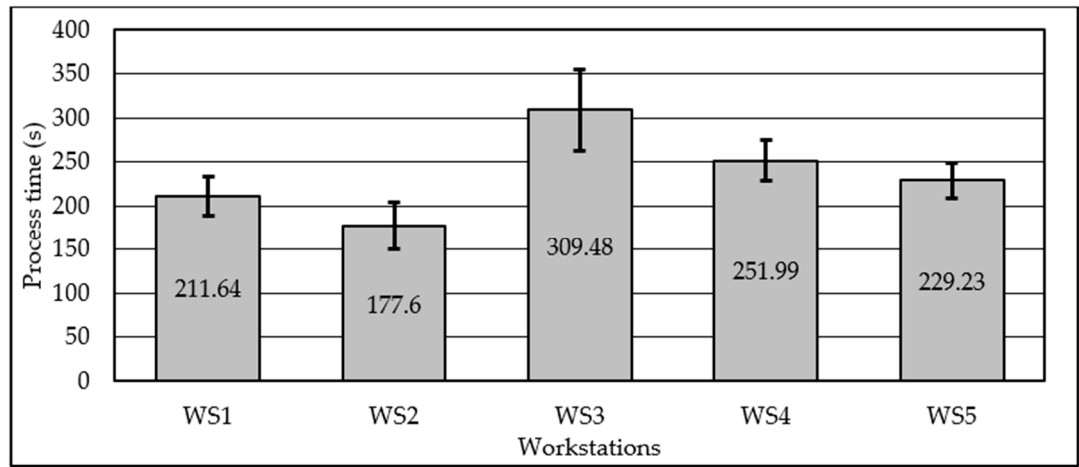

**Figure 8.** Process times and standard deviations in the second experimentation iteration.

The I4.0 system was functioning faultlessly. However, the workers did some mistakes using it, so a total of 14 readings were useless (from total 120). The control stopwatches data were used instead to gather process times.

The errors that workers did are listed according to their number of appearances:

- Worker forgot to complete the process when finished by pressing pushbutton (four appearances).
- Worker did unsuccessful RFID reading by reading the RFID tag too fast (three appearances).
- Worker did not use process abnormality counters (Z, T, and M) and therefore, normal process time was increased abnormally (three appearances).
- Worker did not check all pages of instruction, and therefore, inserted wrong parts (two appearances).
- Worker did not check the production plan, then assembled the wrong type of gearbox (one appearance).
- Worker completed the process by pushbutton immediately after RFID reading (one appearance).

I4.0 system functionality was not compromised at any moment, but workers had to be more focused on display and to check regularly if content shown was aligned to expected. This was emphasized in moments where information flow from human to RFID system take part, i.e., in reading the RFID tag and completing the process with the pushbutton.

As shown in Table 4, the worker on WS3 had many more issues during the assembly process than in the first iteration (Z and T abnormalities readings). The reallocation of assembly steps initiated a new approach in gear lever mechanism assembly. In two moments of the assembly activities, special skills were requested from worker to insert parts while having very small tolerances for insertion. This very complex activity had a high assembly process time and standard deviation. WS3, therefore, became the bottleneck. As a worker on bottleneck process (WS3) do not have idle time, activities performed in the first iteration by the worker on WS3 did not took part, which was another reason why process time on WS3 was considerably longer than predicted in Table 3. Due to the bottleneck

process on WS3, unfinished gearboxes occupied conveyor, which, therefore, enable some idle time on WS1 and WS2. This idle time was thus used for improvements by workers on WS1 and WS2, which reduced respective processes times.

Prior to the third iteration of experimentation, the new observations by analysts should be included. Assembly steps process times on WS1 to WS3 were found different from those gained in the first iteration, due to aforementioned reasons. Nevertheless, the assembly steps in WS4 and WS5 showed negligible differences in process times. Lower accessibility and more complex manipulation needed on WS3 changed assembly steps times significantly compared to those gained in the first iteration. There was a noticeable reduction in total processing time, in comparison to total processing time calculated in the first iteration, which led to reduced ideal overall cycle time (Equation (2)).

$$\bar{c}_t = \frac{\sum\limits_{i=1}^{n} t_{e,i}}{N_{WS}} = \frac{1179.94}{5} = 235.99 \text{ (s)},\tag{2}$$

For the second iteration, the same spreadsheet simulation approach was used. In simulated iteration when assembly line overall cycle time was increased to 255 s, all assembly steps were allocated in only five WSs and was considered as successfully performed balancing process. It was found that only one reallocation of assembly step had to be performed (Table 5).

**Table 5.** Reallocation of assembly steps in balancing procedure after the second iteration.

|  |  | WS1 | WS2 | WS3 | WS4 | WS5 |
|---|---|---|---|---|---|---|
| | The second iteration $t_{e,i}$ (s) | 211.64 | 177.60 | 309.48 | 251.99 | 229.23 |
| Task reallocation | Description: YS0901, YS0902, and YS0904(1) from WS3 to WS2 | - | +55.37 | −55.37 | - | - |
| | Predicted $t_{e,i}$ (s) | 211.64 | 232.97 | 254.11 | 251.99 | 229.23 |
| | Predicted deviation from $\bar{c}_t$ (s) | −24.35 | −3.05 | 18.12 | 16.00 | −6.76 |

After the reorganization of the assembly line and update of instructions pages, the third iteration of experimentation run took place. Workers were trained intensively about how the RFID system works and how it will respond in some boundary conditions.

### 2.3.3. I4.0 Approach for the Third Iteration

In the third iteration, the experiments were performed using "I4.0 approach" for the data gathering process. Analysts were checking cycle times by control stopwatches. Total 24 gearboxes were assembled. The results are presented in Table 6. Average process times with error bars presenting standard deviations are shown in Figure 9.

**Table 6.** Experimentation results for the third iteration.

|  | WS1 | WS2 | WS3 | WS4 | WS5 |
|---|---|---|---|---|---|
| Average process time $t_{e,i}$ (s) | 214.97 | 245.3 | 267.50 | 247.32 | 226.18 |
| Standard deviation of process time $\sigma_{te,i}$ (s) | 19.44 | 29.12 | 39.16 | 21.931 | 26.45 |
| Abnormality Z occurrences | 0 | 0 | 3 | 0 | 1 |
| Abnormality Z average time (s) | - | - | 28.16 | - | 17.31 |
| Abnormality T occurrences | 0 | 2 | 0 | 0 | 0 |
| Abnormality T average time (s) | - | 72.12 | - | - | - |

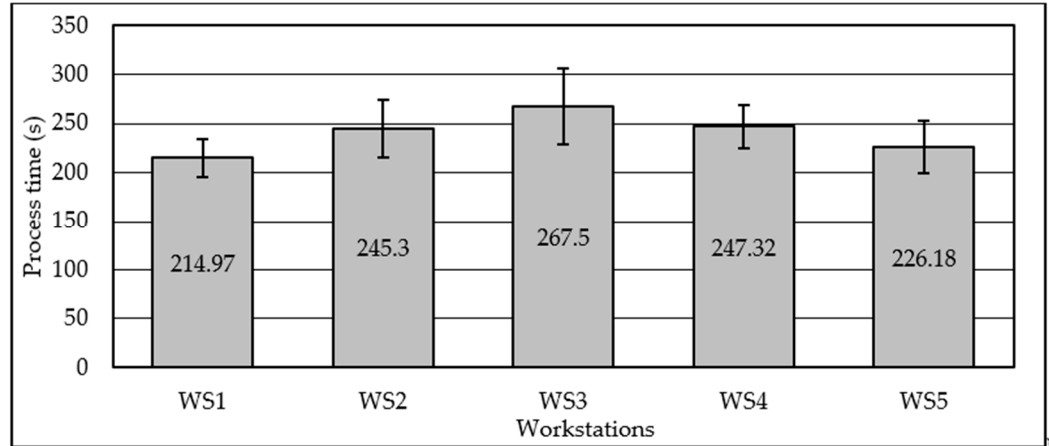

**Figure 9.** Process times and standard deviations in the third experimentation iteration.

The ideal overall cycle time after the third iteration is calculated in Equation (3):

$$\bar{c}_t = \frac{\sum\limits_{i=1}^{n} t_{e,i}}{N_{WS}} = \frac{1201.27}{5} = 240.25 \text{ (s)}, \tag{3}$$

Therefore, in Table 7, deviation from ideal overall cycle time is shown for all WSs.

**Table 7.** Deviation from ideal overall cycle time after the third iteration.

|  | WS1 | WS2 | WS3 | WS4 | WS5 |
|---|---|---|---|---|---|
| Initial $t_{e,i}$ (s) | 214.97 | 245.30 | 267.50 | 247.32 | 226.18 |
| Deviation from $\bar{c}_t$ (s) | −25.28 | 5.05 | 27.25 | 7.07 | −14.07 |

It can be seen that process times were well balanced to some extent. Most of the process abnormalities were recorded in WS3, which, together with high process time standard deviation, presented a clear indicator for the necessity to improve WS3 work procedures, tools, or layout. For manual assembly, acquiring an even better assembly line balancing was not required in most cases, especially when WSs' process times standard deviations were significant and overcame deviation of mean process time from ideal overall cycle time on most of the WSs.

## 3. Discussion

Assembly line balancing process was carried out through the three iterations. In the first iteration, previously allocated assembly tasks were improved with the introduction of electric screwdrivers on all WSs and gravity conveyer for heavy uppers casing on WS5. Introduction of low-volume/high-variety production, i.e., a total of six product types were performed. The experimentation in "Manual approach" resulted with process times on every workstation, as well as the assembly steps process times needed for balancing purposes. The assembly line was not well balanced after the first iteration. Using LCR and spreadsheet simulation, work elements were reallocated in a way that predicted process times on every workstation comply with ideal tact time with appropriate small deviations. For the second iteration, "I4.0 approach" was utilized. Experimentation in the second iteration had shown that the elongation of process time on WS3 triggered with the requirement for assembly with narrow tolerances. Additionally, observations about I4.0 equipment utilization were collected. The third iteration of balancing procedure and experimentation for prediction validation was performed utilizing "I4.0 approach," in order to further balance assembly line with satisfactory deviations of WSs' process times from ideal tact time.

Throughout the assembly line balancing process, I4.0 equipment utilized shown its operational advantages as well as significant drawbacks. As presented, I4.0 RFID system is not only used for assembly process time gathering. I4.0 related equipment triggered server to display respective instruction to workers on interactive touch LCD displays, related to gearbox currently being assembled. Input buttons, defining the end of the assembly process, switching the instructions pages, activating the counters for process abnormalities, and enabling workers to temporarily stop all the counters, give I4.0 RFID systems almost all features previously used in "Manual approach." Drawbacks found will drive the further development of this system and can be found in the following list, together with future activities planned by the authors.

1. I4.0 system cannot make the test at any point of time to confirm the complement of currently assembled product to those being planned for assembly. The assignment of an RFID tag on the product itself could partly solve this problem. The unique RFID tag could be used subsequently in applying individual information to the gearbox which could be useful in the whole supply chain and later in the exploitation phase of the product. Due to the complex geometry of gearbox chassis, and differences in size and shape of different gearboxes, RFID antenna should be portable and not installed on a fixed position on the assembly line. This would increase the workers' activities which is another drawback of I4.0.

2. I4.0 system cannot recognize which page of instructions to show in which moment of the assembly process. Listing using touch screen shown problems, due to oiled fingers on which touch LCD screen do not respond on many occasions. Solution with one standalone pushbutton is not satisfactory as pages are shown in a circular manner. The solution that is planned is installing forward and backward buttons instead. Development of video clips could also make improvement in instruction presentation, but more complex user activities could be required, which is a new drawback of I4.0.

3. I4.0 system uses touch LCD display for most of the inputs from the worker. Due to oiled fingers, many touches are not registered by the resistive touch screen. Installation of keyboard with physical pushbuttons could resolve this issue. The drawback will be the lower level of intuitiveness or reduced user-friendly interface experience.

4. I4.0 system cannot make quality control at any point of assembly, as it is not equipped with sensors enabling it. Solutions can be found in the installation of different sensors, that requires large effort in programming the system. This reduces system robustness and flexibility to new products introduction. Ultimate solution can be found in machine vision, which presents one of the future efforts in subsequent improvement of assembly line using I4.0 related equipment.

## 4. Conclusions

The assembly line balancing procedure resulted in appropriate evenly distributed workloads on total five WSs, with observed smooth material flow achieved throughout the assembly line. The material flow very close to one-piece-flow is observed after the third assembly balancing process iteration and experimentation. No significant inventories were generated between working places during experimentation runs. After the first iteration of ALB, workload of the least utilized WS3 process was only 37.5% of the most utilized (bottleneck) process WS4. After the third iteration of ALB, workload of the least utilized WS1 was 80.4% of the most utilized (bottleneck) process WS3. Nevertheless, process abnormalities recorded define needs for improvements in both assembly processes and logistics, to reduce standard deviations of process times. The I4.0 related equipment embedded in the assembly line showed many advantages in information communication to workers and in the gathering of process times. However, on the other side, I4.0 RFID system could not be used for line balancing purposes as one cannot distinct feasible assembly steps and collect corresponding process times. During I4.0 operation time on total 48 processes, there were at least six types of misuse or faults done by workers directly related to the usage of I4.0 system. Although workers were trained well, on about 6% of

processes, I4.0 RFID system failed to monitor or support processes. Additional 6.3 s on average is used within process time on every WS to perform and check I4.0 system related activities. Nevertheless, increase of information flow speed, reduction of paper sheet consumption, reduction of necessary working space for administrative activities, and simplification of data post-processing showed the unsurpassed advantage of "I4.0 approach" in comparison to "Manual approach."

The future work, besides aforementioned in the previous research, will be focused on enabling real-time monitoring of overall equipment effectiveness (OEE) on I4.0 system. For the manual assembly line, layout reorganization, human–robot collaboration introduction, and machine vision introduction will be considered.

**Author Contributions:** Conceptualization, N.G. and S.K.; Formal analysis, B.S.; Funding acquisition, B.S.; Investigation, N.G.; Methodology, B.S. and A.A.; Supervision, S.K.; Validation, S.K.; Visualization, A.A.; Writing—original draft, N.G.; Writing—review & editing, A.A. All authors have read and agreed to the published version of the manuscript.

**Funding:** This research was funded by King Saud University through researchers supporting project number (RSP-2020/145). And the APC was funded by King Saud University through researchers supporting project number (RSP-2020/145).

**Acknowledgments:** This study received funding from King Saud University through researchers supporting project number (RSP-2020/145).

**Conflicts of Interest:** The authors declare no conflict of interest.

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
