# Peer review of "Utilization of Industry 4.0 Related Equipment in Assembly Line Balancing Procedure"

_processes, doi:10.3390/pr8070864_

Round 1
Reviewer 1 Report
The manuscript titled „Utilization of Industry 4.0 related equipment in assembly line balancing procedure” is a very interesting source of information about the balancing procedure for the assembly of complex components (in this case, gearbox). The manuscript deals with the role of Industry 4.0 related equipment in assembly line balancing process. The assembly process is performed in the Learning Factory environment, and it is considered as very close to real industry process.
The manuscript presents the original research results. The content of the manuscript is compact and correct, and the paper has a proper structure. Both the subject matter and the methodology of conducting experiments are described very well - the research is conducted in a logical and consistent manner. The abstract contains all the most important information - purpose, scope and findings. The Introduction section has an adequate informational level and is an interesting introduction to the topic. References selected correctly. Conclusions have been drawn correctly.
Nevertheless, I note some shortcomings:
- please check the format and citation compatibility with editorial guidelines,
- figure 1 and 2 - a legend of Y/S; A/Z; S/N should appear in the caption of the figure or in the drawing itself - despite that the description is given in the text, but the lack of a legend makes it difficult to understand the assembly diagram
- figure 3 - on what basis „possible relative order of insertion” was determined
- figures 4 are not legible, absolutely, need to be improve
- if the author gives the results of the measurement to two decimal places (hundredths), should apply it to all results for this variable (table 1, column WS1), additionaly the value of the standard deviation should have the same accuracy as the measured variable
- line 258 – Authors write „This effect is not mentioned in many publications in found literature” (any or many?) - Explain the effect (what the authors meant). In the global databases it can by find numerous publications discussing the techniques of assembly processes balancing. I can't find out what phenomenon the Authors are discussing.
An interesting continuation or development of the presented research results would be to present the variability of the OEE indicator (for individual WSs), which is the main determinant of changes in the process. Graphic presentation of changes in the OEE indicator would give a very good definition of the assembly line balancing procedure is performed in “Manual approach” and “I4.0 approach” (this is a proposal for further study).
In my opinion, the subject of the manuscript is very timely and widely described in scientific journals, which is why the literature review seems to be poor
The comments are only constructive criticism. I believe that manuscript meets the requirement for scientific papers published in the Processes Journal, I request its publication after minor correction.
Author Response
Dear reviewer,
please, see attached word file.
Thank you for your comments and requests for improvement,
Best regards,
Authors

Reviewer 2 Report
Dear authors, the adoption of Industry 4.0 technologies in manufacturing processes is a hot topic in the extant literature and papers focusing on that are always welcome. However, there are several issues to be improved before publication:
1) Abstract is too unbalanced on results and lacks of clarity in terms of general context, main research gaps assessed by the paper, research method adopted, future trends.
2) Introduction must better evidence what are the existing issues to be solved. In addition, a summary of the sections consituting the paper must be added at the end of section 1.
3) The paper does not seems to follow the standard format of the journal. Please, re-structure it.
4) A section dedicated to a literature review of existing works in the field must be added
5) A section dedicated to the research methodology must be added
6) Conclusions must be improved by evidencing main results, limitations, theoretical and managerial contributions
7) References must be improved
Author Response

(The authors gave the same response as above.)
